# Morphological Stages of Mitochondrial Vacuolar Degeneration in Phenylephrine-Stressed Cardiac Myocytes and in Animal Models and Human Heart Failure

**DOI:** 10.3390/medicina55060239

**Published:** 2019-06-03

**Authors:** Antoine H. Chaanine

**Affiliations:** Division of cardiovascular diseases, Mayo Clinic, Rochester, MN 55902, USA; chaanine78@gmail.com; Tel.: +1-646-648-1874

**Keywords:** heart failure, oxidative stress, mitochondria, dynamics, morphology, vacuolar degeneration

## Abstract

*Background and objectives*: Derangements in mitochondrial integrity and function constitute an important pathophysiological feature in the pathogenesis of heart failure (HF) and play an important role in myocardial remodeling and systolic dysfunction. In systolic HF, we and others have shown an imbalance in mitochondrial dynamics toward mitochondrial fission and fragmentation with evidence of mitophagy, mitochondrial vacuolar degeneration, and impairment in mitochondrial oxidative capacity. The morphological stages of mitochondrial vacuolar degeneration have not been defined. We sought to elucidate the progressive stages of mitochondrial vacuolar degeneration, which would serve as a measure to define, morphologically, the severity of mitochondrial damage. *Materials and Methods*: Transmission electron microscopy was used to study mitochondrial morphology and pathology in phenylephrine-stressed cardiac myocytes in vitro and in left ventricular myocardium from a rat model of pressure overload induced systolic dysfunction and from patients with systolic HF. *Results*: In phenylephrine-stressed cardiomyocytes for two hours, alterations in mitochondrial cristae morphology (Stage A) and loss and dissolution of mitochondrial cristae in one (Stage B) or multiple (early Stage B→C) mitochondrion area(s) were evident in the earliest stages of mitochondrial vacuolar degeneration. Mitochondrial swelling and progressive dissolution of mitochondrial cristae (advanced Stage B→C), followed by complete loss and dissolution of mitochondrial cristae and permeabilization and destruction of inner mitochondrial membrane (Stage C) then outer mitochondrial membrane rupture (Stage D) constituted advanced stages of mitochondrial vacuolar degeneration. Similar morphological changes in mitochondrial vacuolar degeneration were seen in vivo in animal models and in patients with systolic HF; where about 60–70% of the mitochondria are mainly observed in stages B→C and fewer in stages C and D. *Conclusion*: Mitochondrial vacuolar degeneration is a prominent mitochondrial morphological feature seen in HF. Defining the progressive stages of mitochondrial vacuolar degeneration would serve as a measure to assess morphologically the severity of mitochondrial damage.

## 1. Introduction

Mitochondria are organelles where energy production takes place in the cell, in the form of adenosine triphosphate essential to maintain cellular function, through a process of oxidative phosphorylation (OXPHOS) and mitochondrial respiration. Thus, they are so called the powerhouse of the cell and are quintessential in organs of high energetic demand, such as the heart, brain, and skeletal muscle [1]. These organelles are mostly abundant and have the densest cristae in the heart and constitute about 30% of the cardiac myocyte volume [1,2]. Beside their role in energy production, mitochondria are arranged in a complex fashion to lie at a very close proximity with the adjacent T-tubules, longitudinal sarcoplasmic reticulum (SR), and the sarcomeres in cardiac myocytes [3]. These aforementioned organelles cross talk with each other via calcium signaling. For instance, tethering complexes exist between the longitudinal SR and the adjacent intermyofibrillar mitochondria that play a role in calcium influx and efflux in and out of the mitochondria into the longitudinal SR on a beat-to-beat basis [4,5]. Under physiological conditions, the intensity of calcium influx into the mitochondria is a critical stimulus, at least in the cardiac myocyte, for ATP production but at the expense of reactive oxygen species (ROS) production [6,7,8]. Moreover, they are such dynamic organelles that undergo cycles of mitochondrial fusion and fission, critically balanced and essential for mitochondrial biogenesis and mitochondrial quality control. Mitochondria that divide and are unable to refuse, due to loss in their membrane potential, are targeted for elimination through mitophagy, essential for the maintenance of mitochondrial quality control and prevention of mitochondrial induced apoptosis and cell death [9]. Thus, they play a critical role in promoting cell survival or executing cell death in response to external stimuli and the activation of signal transduction pathways promoting cell survival or cell death.

Heart failure (HF) is a disease state of heightened oxidative stress [10] where ROS and non-ROS induced activation of detrimental signal transduction pathways results in derangement in mitochondrial morphology and function leading to mitochondrial dysfunction and metabolic remodeling as well as altered mitochondrial dynamics, in favor of mitochondrial fission and fragmentation, and decreased mitochondrial biogenesis [11,12,13,14]. These aforementioned derangements in mitochondrial integrity and homeostasis play a central role in the pathogenesis of HF and adversely affect myocardial remodeling [11,12,13,15,16]. Highlighting these signal transduction pathways is out of the scope of the manuscript. The scope of this manuscript is to characterize and define morphologically, utilizing transmission electron microscopy, the progressive stages of mitochondrial vacuolar degeneration. Thus, we present and define for the first time in the literature the progressive stages of mitochondrial vacuolar degeneration in phenylephrine (PE)-stressed cardiac myocytes, in vitro, and in rat models of pressure overload (PO) induced systolic dysfunction as well as in human systolic heart failure; therefore, in conditions of heightened oxidative stress. The characterization of the different stages of mitochondrial vacuolar degeneration is based on our experience and observation from a library of transmission electron microscopy photomicrographs available to us. The identification and classification of the progressive stages of mitochondrial vacuolar degeneration are important in the sense that they will serve as a measure to delineate morphologically, the degree of mitochondrial damage in cells subjected to external apoptotic stimuli, and in disease states in vivo.

## 2. Materials and Methods

The data, analytic methods, and study materials will not be made available to other researchers for purposes of reproducing the results or replicating the procedure.

### 2.1. Isolation of Cardiac Myocytes

The isolation of adult cardiac myocytes (ACMs) and their transfection with adenoviruses encoding BNIP3 or constitutively active FOXO3a, or stressing them with PE was described elsewhere [17]. Briefly, rat ACMs were isolated from male Sprague-Dawley rats weighing (250–350 g) using the langendorff isolation system. The animals were given sodium heparin (100 U) and sodium pentobarbital (15 mg) for anesthesia by intraperitoneal injection, prior to cervical dislocation. Then the chest was opened and the heart was removed and immediately cannulated as quickly as possible and perfused using Krebs-Henseleit buffer containing (in mmol/L): 118 NaCl, 4.7 KCl, 1.25 KH2PO4, 1.3 MgSO4, 10 glucose, and 10 HEPES (pH 7.4) for 4 min at 35 °C. All reagents were obtained from Sigma (Sigma, St. Louis, MO, USA). The buffer was pumped through the heart by means of peristaltic pump at a rate of 10 mL/min. Then the heart was perfused with Krebs–Henseleit buffer with low Ca^2+^ content (<0.25 mmol/L), pH 7.4, with collagenase (2 mg/mL, Worthington, Lakewood, NJ, USA) for 15–20 min. After 15 min, the heart was removed from the apparatus, and the ventricles were separated below the atrioventricular junction, then the heart was cut into 1–2 mm^2^ fragments and carefully resuspended for 1 min with a large automated pipette. The cells were filtered through a nylon mesh cell strainer, pore size 100 μm (San Jose, CA, USA), and allowed to settle by gravity for 5 min. The cell pellet was subsequently suspended in an incubation buffer (in mmol/L: 118 NaCl, 25 NaHCO3, 4.7 KCl, 1.2 KH2PO4, 1.2 MgSO4, 10 glucose, 30 N-2 hydroxyethylpiperazine-N-2 ethanesulfonic acid (HEPES), 60 taurine, 20 creatine, 1% bovine serum albumin, vitamins, and amino acids (Sigma, St. Louis, MO, USA) at pH 7.4, 37 °C. The calcium concentration was gradually increased to 1.2 mmol/L over 20 min. ACMs were then re-suspended in Medium M199 containing 10-units/mL penicillin, 10 μg/mL streptomycin, 5% bovine serum albumin and 100 nmol/L insulin (Sigma, St. Louis, MO, USA). Using this method, each heart yielded ≈7 × 106 rod-shaped ACMs with viability greater than 80%. The ACMs were placed on a laminin coated culture dishes in full nutrient M199 medium as mentioned above and were allowed to attach for 1 h in humidified 5% CO2, 95% air at 37 °C and then washed once to remove unattached cells. Then the ACMs were stressed with PE (Sigma, St. Louis, MO, USA) for two hours at a concentration of 10 μmol/L. Then cells were fixed and processed for transmission electron microscopy. Presented data were obtained from at least three independent experiments. The transfection of the ACMs with adenovirus encoding BNIP3 or constitutively active FOXO3a is presented and described in detail elsewhere [17]. Presented data were obtained from at least three independent experiments.

### 2.2. Experimental Model of Ascending Aortic Banding and Retrieval of Left Ventricular Subepicardial Biopsy

All procedures involving the handling of animals were approved by the Animal Care and Use Committee (IRB approval number is: 15-008625) of the Mayo Clinic and adhered to the Guide for the Care and Use of Laboratory Animals published by the National Institutes of Health. Sprague-Dawley rats weighing 180–200 g underwent ascending aortic banding (AAB). The animals were sedated by the intraperitoneal administration of ketamine (65 mg/kg) plus xylazine (5 mg/kg) and intubated using a 16-gauge catheter and mechanically ventilated with tidal volumes of 2 mL at 50 cycles/min and FIO2 of 21%. A 1 cm incision was made in the right axilla and the thoracic cage was approached at the level of the second intercostal space. The thymus gland was dissected, then the underlying ascending aorta was separated from the superior vena cava and a 1mm (2 mm^2^ area) vascular clip was placed around the ascending aorta, right before the right brachiocephalic artery. Buprenorphine SR (0.6 mg/kg) was administered as a single dose subcutaneously for analgesia after the surgery. PO developed right after the placement of the vascular clip. The size of the clip was adjusted for the generation of a model of moderate left ventricular (LV) remodeling and systolic dysfunction as published elsewhere [13]. The animals were screened by echocardiography and were included, at eight weeks after AAB, if there was evidence of increased LV end-diastolic (LVEDV) and end-systolic (LVESV) volumes (LVEDV ≥ 600 μL and LVESV ≥ 120 μL at week eight after AAB compared to LVEDV ≅ 300 μL and LVESV ≅ 28 μL at week 3 after AAB). Sham operated animals underwent the same procedure but without the placement of a vascular clip (LVEDV ≅ 540 μL and LVESV ≅ 95 μL at week 8 from the sham procedure). Left ventricular (LV) myocardium tissue from animals with PO-induced systolic HF was obtained to study mitochondrial ultrastructural changes. The number of animals studied is *n* = 5 for the sham group and *n* = 7 for the HF group.

### 2.3. Retrieval of LV Subepicardial Biopsy

Mayo Institutional Review Board approval was obtained and all participants provided written informed consent. The detailed description of the retrieval of LV subepicardial biopsy and its dissection from patients with systolic HF is described elsewhere [15]. Two control surgical LV subepicardial biopsies obtained from two patients with normal LV size and function, who underwent cardiothoracic surgery for ascending aortic root replacement without aortic valve pathology and cardiac myxoma resection, respectively, were studied ultrastructurally utilizing transmission electron microscopy. Thirteen surgical LV subepicardial biopsies were studied ultrastructurally that were obtained from patients with systolic HF who underwent cardiothoracic surgery for aortic valve replacement due to severe aortic stenosis (*n* = 4), coronary artery bypass grafting for 3-vessel coronary artery disease (*n* = 5), and left ventricular assist device implantation for advanced HF (*n* = 4).

### 2.4. Transmission Electron Microscopy

Cardiomyocytes and fractions (1 mm^3^ each), from LV myocardium of animals with PO-induced systolic dysfunction and from LV subepicardial biopsy from patients with systolic HF, were pre-fixed in a solution of Trump’s fixative overnight at 4 °C, post-fixed in 1% osmium tetroxide (OsO4), dehydrated in an ascending series of alcohols, and embedded in epoxy resin, as previously described [18]. Ultrathin sections were stained with uranylacetate and lead citrate. The samples were viewed with the JEOL 1400 transmission electron microscope (JEOL Ltd., Akishima, Tokyo, Japan). Transmission electron microscopy photomicrographs were taken at 5k, 12k, 20k, and 40k × magnification. The quantification of the different stages of mitochondrial vacuolar degeneration was performed in 12k × magnified images and was presented as the percentage of total mitochondria per magnified field (12k ×) of the transmission electron photomicrograph.

### 2.5. Statistical Analysis

Data are presented as box and whisker plots; lines showing median, 25th and 75th percentiles with whiskers showing minimum and maximum values. Two-way ANOVA with correction for multiple comparisons using the Bonferroni method was used to assess for statistical significance in each stage of mitochondrial vacuolar degeneration within the different groups. For all statistical analyses, the interaction P value was significant, <0.001, and a *p* value of <0.05 was considered statistically significant. Analysis was performed in GraphPad Prism (version 8; GraphPad Software Inc., La Jolla, CA, USA).

## 3. Results

### 3.1. Morphological Stages of Mitochondrial Vacuolar Degeneration in Phenylephrine-Stressed Cardiomyocytes In Vitro

We assessed mitochondrial morphological changes in PE-stressed cardiac myocytes for two hours in vitro using transmission electron microscopy. At two hours, there was evidence of mitochondrial fission and fragmentation as well as mitophagy, Figure 1.

Also, there were mitochondria at different stages of vacuolar degeneration. The early stages of mitochondrial vacuolar degeneration are defined as stage A, sage B, and early stage B→C. In stage A, mitochondrial cristae become concentrically arranged in an onion-like pattern, red dots in Figure 2 stage A-zoomed photomicrograph, with widened space between them, blue bars in Figure 2 stage A-zoomed photomicrograph. These early conformational changes in mitochondrial cristae morphology and the widened space between them signal early loss in mitochondrial cristae and probably early mitochondrial dysfunction. The second stage of mitochondrial vacuolar degeneration, stage B, is defined as a focal loss and dissolution of mitochondrial cristae in one mitochondrion area. The third stage is a transition stage between stages B and C and is defined as stage B→C, where there is multifocal loss and dissolution of mitochondrial cristae per mitochondrion, Figure 2 Stages B and B→C. In the advanced stages of mitochondrial vacuolar degeneration there is evidence of mitochondrial swelling. These advanced stages are defined as advanced stage B→C, stage C, and stages D and D^+^, the remaining photomicrographs in Figure 2. In advanced stage B→C, there is a significant dissolution of a great proportion of mitochondrion cristae. The inner mitochondrial membrane (IMM) and the outer mitochondrial membrane (OMM) remain intact. Stage C highlights complete dissolution of mitochondrial cristae and the IMM. The OMM remains intact. Stage D highlights findings in stage C plus the rupture of the OMM. Stage D+ is the stage where numerous adjacent mitochondria had undergone complete dissolution of mitochondrial cristae, IMM and OMM.

In PE-induced apoptotic cardiac myocytes, numerous mitochondria are observed at an advanced stage of vacuolar degeneration, mainly stage B→C and stages C and D, Figure 3 A,B. Note the presence of an apoptotic-fragmented nucleus, arrows in Figure 3A-zoomed photomicrograph, with evidence of chromatin condensation into a ring-like pattern at the nuclear periphery in one of the fragmented nuclei, arrowhead in Figure 3A-zoomed photomicrograph. Also, note the presence of apoptotic bodies containing numerous mitochondria, arrows in Figure 3B-zoomed photomicrograph. These morphological changes of mitochondrial vacuolar degeneration are seen in isolated cardiac myocytes transfected with an adenovirus encoding for the mitochondrial death and mitophagy marker, BNIP3, and an adenovirus encoding for a constitutively active FOXO3a, which is a transcription factor that has been shown to induce BNIP3 gene expression in cardiac myocytes, Figure 4A and Figure 4B-blue arrows, confirming the presence of the described mitochondrial morphological changes in apoptotic conditions other than PE. The attributed mechanism will be discussed later in the discussion section. Also, note the presence of autophagosomes containing fragmented mitochondria, Figure 4A,B-black arrows. The zoomed photomicrograph in Figure 4A is showing an autophagosome containing fragmented mitochondrion fusing with a lysosome.

### 3.2. Morphological Stages of Mitochondrial Vacuolar Degeneration in a Rat Model of Pressure Overload-Induced Systolic Dysfunction and in Human Systolic Heart Failure

Similar stages of mitochondrial vacuolar degeneration are observed in vivo except that morphological changes in mitochondrial cristae observed in stage A in vitro, are not seen in vivo. In stage B, there is focal loss and dissolution of mitochondrial cristae in one mitochondrion area, Figure 5 stage B-zoomed photomicrograph-blue arrow and Figure 6 stage B-zoomed photomicrograph. Conformational morphological changes in mitochondrial cristae show that they are uniquely arranged in a horizontal stacked-up pattern, rather than being dense and folded. Also, there is evidence of mitochondria undergoing fission, early (Figure 5 stage B-blue line surrounding mitochondrion in left side of photomicrograph), mid and late (Figure 5 stage B-blue line surrounding mitochondrion in right lower and right upper side of photomicrograph, respectively). In early stage B→C, slight mitochondrial swelling is seen with multifocal loss and dissolution of mitochondrial cristae per mitochondrion area. Mitochondrial swelling becomes more evident in advanced stage B→C with more pronounced loss and dissolution of mitochondrial cristae per mitochondrion area. The IMM and the OMM remain intact in stage B→C, Figure 5 and Figure 6 stage B→C. Similar patterns of mitochondrial vacuolar degeneration are seen in stages C, D, and D^+^ in vivo, as described above in vitro, Figure 5 and Figure 6 Stages C, D, and D^+^.

The quantification of the different stages of mitochondrial vacuolar degeneration in control and PE-stressed and PE-apoptotic ACMs is presented in Figure 7A along with a representative transmission electron photomicrograph of myocardium from a control ACM. The quantified data in Rat-HF and human-HFrEF are presented in Figure 7B,C along with a representative transmission electron photomicrograph of a normal LV myocardium. The majority of mitochondria are in stages A, B, and early B→C in PE-stressed ACMs for 2 h, while in PE-apoptotic ACMs the majority of mitochondria are in stage B→C (early, but mainly advanced) and in stages C and D compared to the control ACM, Figure 7A. In rat-HF and human-HFrEF, about 60–70% of mitochondria on average are in stage B→C (early and advanced) and fewer in stages C and D, but significant compared to Normal LV myocardium.

## 4. Discussion

We describe and characterize for the first time the progressive stages of mitochondrial vacuolar degeneration in cardiac myocytes in vitro and in HF in vivo. These morphological changes are seen in conditions of heightened oxidative stress and ROS production, both in vitro and in vivo. A common etiology for mitochondrial vacuolar degeneration and increased mitochondrial ROS production is mitochondrial calcium matrix overload. Increases in mitochondrial calcium content enhances enzymes in Krebs cycle and potentiates the activity of and ROS production from the electron transport chain complexes I and III [19], leading to OXPHOS uncoupling and decreased ATP production [17,20,21]. Simultaneously, increased mitochondrial calcium and ROS is a potent stimulus for the activation of the mitochondrial permeability transition pore (MPTP) [22], leading to the opening of the MPTP and increase in its conductance to small solutes and water. This causes mitochondrial osmotic swelling and eventually leads to IMM permeabilization and the rupture of OMM and release of cytochrome c [23]. Moreover, it has been shown that increased mitochondrial calcium activates mitochondrial proteases [24] and Calpain I [25] leading to enhanced mitochondrial proteolysis and the cleavage and release of the apoptosis-inducing factor, respectively.

We have shown that the activation of the FOXO3a-BNIP3 axis promotes SR-mitochondrial calcium shift and mitochondrial matrix calcium overload and mitochondrial dysfunction in cardiac myocytes, in vitro and in vivo [17,26]. BNIP3 monomer expression significantly increases in PE-stressed cardiomyocytes and in constitutively active FOXO3a expressing cardiomyocytes in vitro [17,26] and in animal models of myocardial remodeling and systolic dysfunction [13,14,17,26,27], in vivo, as well as in human systolic heart failure [15]. Indeed, all stages of mitochondrial vacuolar degeneration are evident in the above aforementioned conditions, with the exception that in vivo, stage A of mitochondrial vacuolar degeneration is not observed. One or very few mitochondria at the advanced stage of vacuolar degeneration could be observed in live, non-apoptotic cardiomyocytes, without the necessary commitment to apoptosis. However, in PE-induced apoptotic cardiac myocytes, numerous mitochondria in advanced stages of vacuolar degeneration are observed. This suggests that the higher the number of mitochondria in the advanced stages of vacuolar degeneration, the higher is the likelihood of the induction of apoptosis, secondary to the profound leakage of cytochrome c and proapototic factors and endonucleases. From the quantification of the different stages of mitochondrial vacuolar degeneration, presented in Figure 7, one would observe that in systolic HF about 60–70% of the mitochondria on average are in stage B→C (early and advanced) and fewer in stages C and D, a disease state associated with mitochondrial dysfunction and enhanced apoptosis [13,15]; this is relatively similar to the trend seen in PE-apoptotic ACM. It remains unclear to what the threshold of mitochondria in advanced stages of vacuolar degeneration would be for apoptosis to take place. Also, at which stage of mitochondrial vacuolar degeneration does mitochondrial damage become irreversible, even if the external stressor is removed, remains an area of uncertainty.

It has been shown that the permeability of the OMM is mandatory for apoptotic cytochrome c release and this has been considered the ‘point of no return’ in the cell death process. However, this theory has been challenged, as there is evidence of partial release of cytochrome c without the necessary activation of caspases and the initiation of the apoptotic process, but was paradoxically shown to promote cell survival [28]. However, at higher concentrations of cytochrome c release is when the caspases are activated promoting the apoptotic cell death [28]. It is very well known that the BCL-2 family proteins regulate OMM permeability and apoptosis [29,30,31,32]. These are divided into three main groups based on their BCL-2 homology domains and function: (a) the anti-apoptotic BCL-2 members, such as BCL-2 and BCL-XL, (b) the pro-apoptotic multidomain members, such as BAX, and (c) the BH3 only proapoptotic members, such as BID and BNIP3. The proapototic members, therefore the multidomain and the BH3 only domain proapoptotic proteins, act synergistically to promote OMM permeabilization and cytochrome c release via distinct mechanisms prior to the initiation of apoptosis. BAX promotes protein-permeable pores in the OMM, without affecting the IMM and mitochondrial matrix or the occurrence of mitochondrial swelling, while BNIP3 promotes the oligomerization of the voltage dependent anion channels (VDAC), mitochondrial calcium overload, MPTP activation, mitochondrial swelling, and OMM rupture and release of cytochrome c [26]. Moreover, VDAC oligomerization serves as a large pore channel for the release of cytochrome c and induction of apoptosis in response to external apoptotic stimuli [33]. Both BAX and BNIP3 expression are increased in animal models and in human HF [14]. In animal models of PO-induced hypertrophy and systolic dysfunction; the ratio of BAX to BCL-2 is increased as early as one week after PO and remains about the same thereafter until the development of HF; whilst, BNIP3 expression increases at 2–3 weeks after PO and peaks at HF development [14], therefore amplifying and potentiating mitochondrial dysfunction and the apoptotic process [13]. Indeed, we found a strong correlation between the intensity of BNIP3 monomer expression and the degree of mitochondrial fragmentation, mitochondrial dysfunction and myocardial remodeling in animal models and in human HF [14,15]. As more mitochondria become dysfunctional and progress into an advanced stage of vacuolar degeneration, the apoptotic process is amplified, leading to myocardial remodeling and decline in cardiac function. This was evident in human HF with preserved ejection versus reduced ejection fraction of similar etiology, whether related to severe aortic stenosis or ischemic heart disease. Patients with HF and reduced ejection fraction had evidence of mitochondrial dysfunction and more pronounced advanced stages of mitochondrial vacuolar degeneration compared with patients with HF and preserved ejection fraction [15].

Derangements in mitochondrial cristae morphology are seen in the earliest stages of mitochondrial vacuolar degeneration. Mitochondrial cristae are arranged in a concentric onion-like pattern in stage A, in vitro, and horizontally in a stacked-up pattern in stage B, in vivo. It would be interesting to elucidate molecular mechanisms responsible for these early morphological changes in mitochondrial cristae and understand whether mitochondrial respiration and oxidative capacity is impaired in the earliest stages of mitochondrial vacuolar degeneration. Obviously, as mitochondrial swelling ensues and with the transition into advanced stages of vacuolar degeneration that mitochondrial oxidative capacity would become impaired, as the mitochondria would have lost most of their cristae. Mitochondrial cristae are deep invaginations of the IMM into the mitochondrial matrix. Thus the IMM is divided into two domains: (a) the inner boundary membrane, which lies adjacent and interior to the OMM and contains protein translocases that import nuclear-encoded proteins into the mitochondria, and (b) the cristae membranes that form invaginations of variable size and shape and are enriched with the electron transport chain complexes I, II, III, IV, and V. Therefore, the cristae membranes are the hub of electron transport and mitochondrial respiration. The connection between cristae and the inner boundary membrane is formed by a homogeneous structure that has been named as crista junction. Crista junctions are narrow, neck-like structures that are characterized by a high degree of negative membrane curvature. They are thought to represent a diffusion barrier for proteins as well as small molecules. In mammalian cells the crista junctions are maintained and stabilized by the mitochondrial contact site and cristae organizing system (MICOS) in association with the dynamin-related GTPase OPA1 [34]. The MICOS complex is formed from a number of proteins, most important of which are Mic60 and Mic10. Mutations or knockdown of the MICOS proteins, ATP synthase, and the dynamin-related GTPase OPA1 are associated with severe defects in cristae morphology leading to a number of cardiovascular and neurodegenerative diseases [34]. For instance, in yeast mutants lacking Atp20 or Atp21, the cristae membranes are deranged and display an onion-like morphology [35]. Whereas, mutations in or knockdown of Mic60, the cristae membranes are arranged in stacks [36,37], and are associated with a decline in mitochondrial oxidative capacity and has been shown to be associated with a number of diseases such as Parkinson’s disease, epilepsy, diabetic cardiomyopathy, and neurodegeneration [34]. Similarly, mutation in OPA1 causes derangement in cristae morphology and mitochondrial oxidative capacity and has been associated with the autosomal dominant optic atrophy and Charcot-Marie-Tooth disease type 2A [34]. Moreover, it has been shown that the transmembrane domain of BNIP3 binds and inhibits OPA1 leading to mitochondrial fragmentation, cristae destruction and release of cytochrome c and apoptosis [38].

Autophagy is a self-eating process where aggregates of proteins or damaged mitochondria are engulfed within double membranes, called autophagosomes, which then fuse with the lysosomes for the degradation of their content. Therefore, the autophagic process is thought to be cell protective by removing damaged mitochondria prior to the initiation of apoptosis. There is a clear evidence of mitochondria-containing autophagosomes in PE-stressed cardiomyocytes and in cardiac myocytes overexpressing BNIP3 or expressing a constitutively active FOXO3a [14,17]. Also, we have shown evidence of autophagosomes fusing with lysosomes in the above conditions in vitro and in animal models and in human HF [14,15,17]. Under physiological conditions, the autophagic process can efficiently eliminate damaged organelles and aggregates of proteins and thus is able to maintain cellular homeostasis and viability. However, in disease states, such as HF, this process is overwhelmed and is unable to keep up with signaling pathways driving mitochondrial dysfunction and remodeling [39]. Moreover the lysosomes become saturated and dysfunctional due to the peroxidation of the lysosomal membrane by ROS, leading to the instability of the lysosomes and to the accumulation of lipofuscin material [40,41]. This phenomenon is also clearly observed in aging [40,41] and in Chloroquine-associated cardiomyopathy [10].

## 5. Conclusions

This study highlights the progressive morphological changes of mitochondrial vacuolar degeneration. The characterization of the variable stages of mitochondrial vacuolar degeneration is important in the sense that it will serve as a measure to assess morphologically the degree of mitochondrial damage in cells subjected to external apoptotic stimuli and in disease states associated with derangements in mitochondrial function and integrity. The progression of mitochondria from an early to a more advanced stage of vacuolar degeneration becomes detrimental as more and more mitochondrial are observed in advanced stages of vacuolar degeneration. Questions remain as to: (1) at which stage of mitochondrial vacuolar degeneration does mitochondrial oxidative capacity become impaired, and (2) at which stage of mitochondrial vacuolar degeneration does mitochondrial damage and dysfunction become irreversible, and (3) what is the threshold of mitochondrial dysfunction and cytochrome c release at which apoptosis is initiated. Also, it would be interesting to understand the molecular mechanisms that govern the morphological changes in cristae morphology during the earliest stages of mitochondrial vacuolar degeneration. These questions may need to be answered in the future.

## Figures and Tables

**Figure 1 medicina-55-00239-f001:**
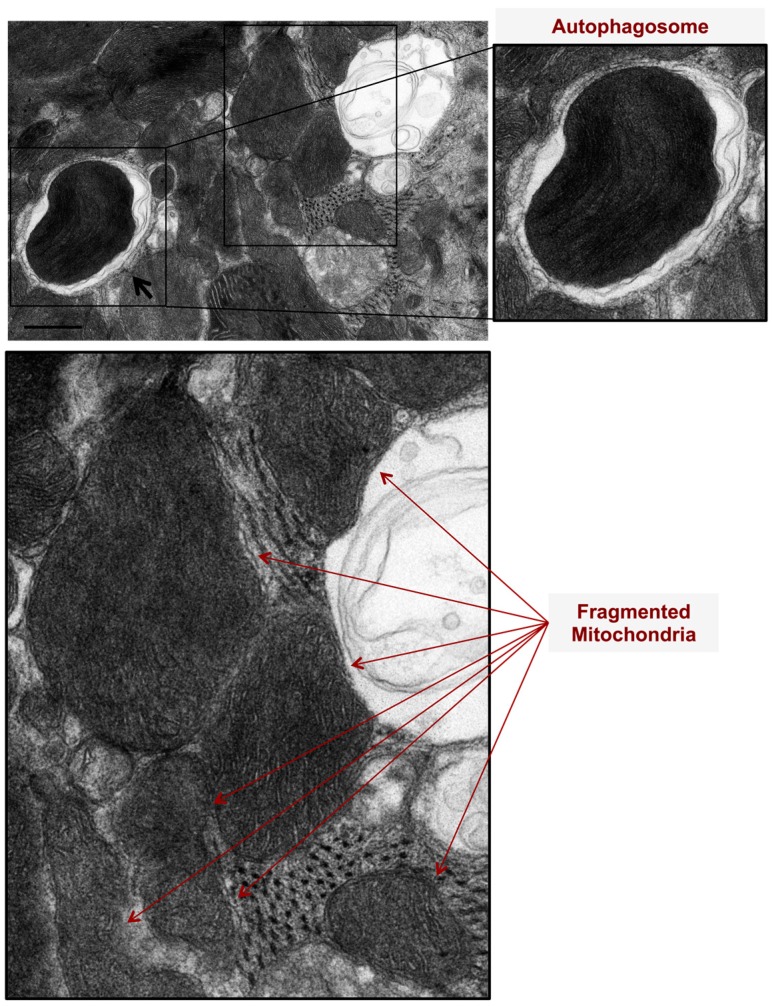
Mitochondrial autophagy (mitophagy) and mitochondrial fragmentation are pathological features evident in phenylephrine-stressed cardiac myocytes, in vitro. Cardiac myocytes were stressed with 10 μM Phenylephrine for 2 h. Note the presence of autophagosomes, double membranes, and surrounding damaged and fragmented mitochondria (black arrows). Also, note the presence of fragmented mitochondria (red arrows). Photomicrograph is 40k × magnified, scale bar 0.5 μm.

**Figure 2 medicina-55-00239-f002:**
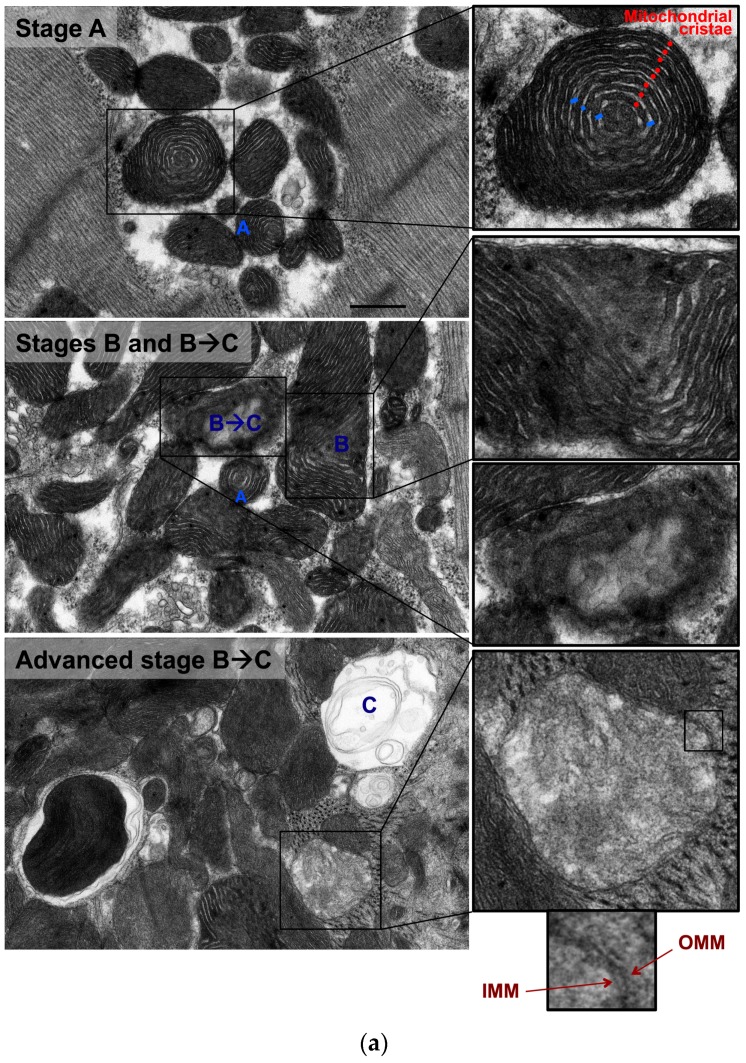
Morphological stages of mitochondrial vacuolar degeneration in phenylephrine-stressed cardiac myocytes in vitro. Stage A: Highlights concentric, onion-like arrangement of mitochondrion cristae (small red dots) with the widening of spaces between adjacent cristae (blue bars). Stage B: Dissolution of mitochondrion cristae in one mitochondrion area. Stage B→C: (**a**) Early dissolution of mitochondrion cristae in multiple mitochondrion areas. (**b**) Advanced stage B→C, mitochondrial swelling with almost complete dissolution of mitochondrial cristae. The inner mitochondrial membrane (IMM) and outer mitochondrial membrane (OMM) remain intact. Stage C: Mitochondrial swelling with the entire dissolution of mitochondrion cristae and the IMM. The OMM remains intact. Stage D: Stage C with an associated rupture of OMM (blue arrows). Stage D^+^: Area(s) where numerous adjacent mitochondria had undergone complete vacuolar degeneration and the entire dissolution of mitochondrial cristae, IMM and OMM. Photomicrographs are 40k × magnified, scale bar 0.5 μm.

**Figure 3 medicina-55-00239-f003:**
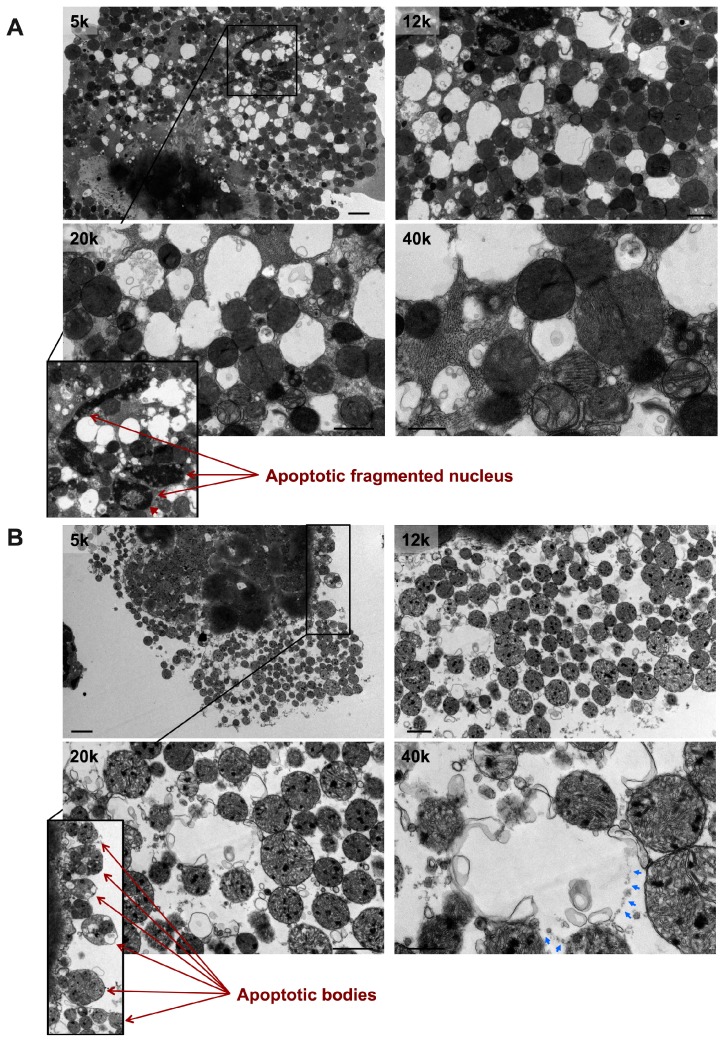
Transmission electron photomicrographs of PE-induced apoptotic cardiac myocytes. Note the presence of a fragmented nucleus (**A**) and apoptotic bodies (**B**), red arrows in A and B; a pathognomonic features of apoptosis. Apoptotic cardiac myocytes show numerous mitochondria undergoing stage C and stage D of vacuolar degeneration with the remainder mitochondria in stage B→C and very few in stage B. Red arrowhead in A pointing at chromatin condensation into a ring-like pattern into the nuclear periphery. Blue arrows in B pointing at sites of OMM rupture of a swollen mitochondrion. Photomicrographs are 5k, 12k, 20k, and 40k × magnified, scale bar 2, 1, 1, and 0.5 μm, respectively.

**Figure 4 medicina-55-00239-f004:**
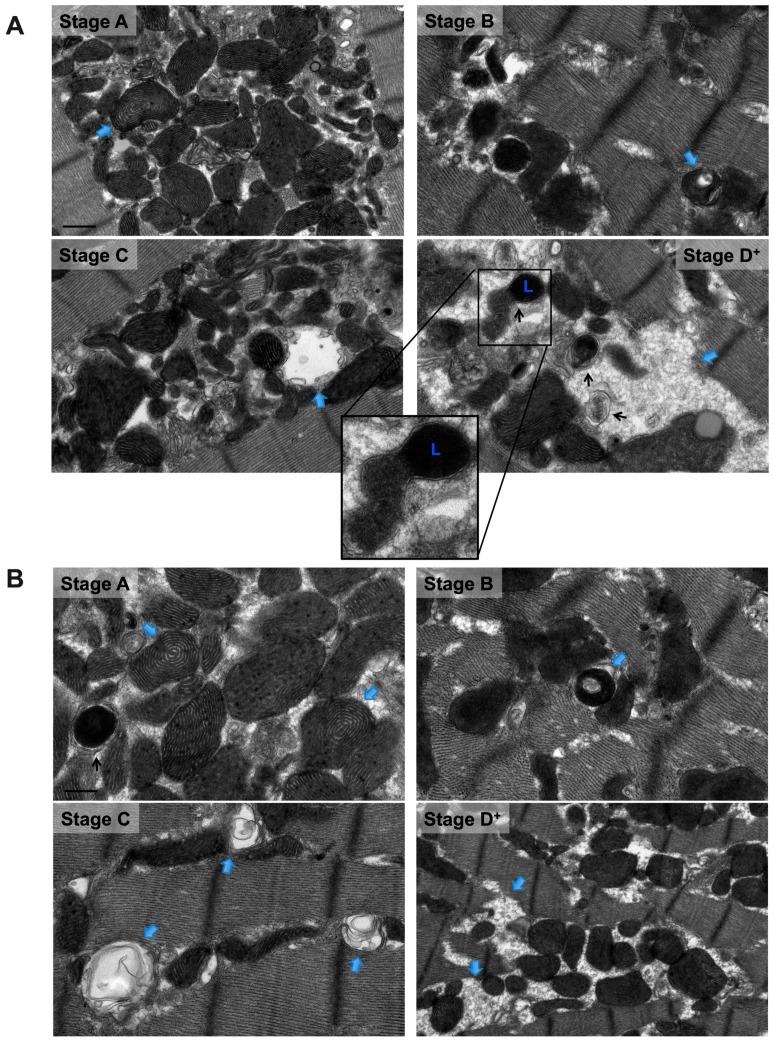
Transmission electron photomicrographs showing similar morphological stages of mitochondrial vacuolar degeneration in cardiac myocytes overexpressing BNIP3 (**A**) and constitutively active FOXO3a (**B**). Blue arrows pointing at mitochondrion at a specific stage of mitochondrial vacuolar degeneration. Black arrows pointing at autophagosomes. Zoomed photomicrograph in A, showing an autophagosome containing a fragmented mitochondrion fusing with a lysosome. L: lysosome. Photomicrographs are 40k × magnified, scale bar 0.5 μm.

**Figure 5 medicina-55-00239-f005:**
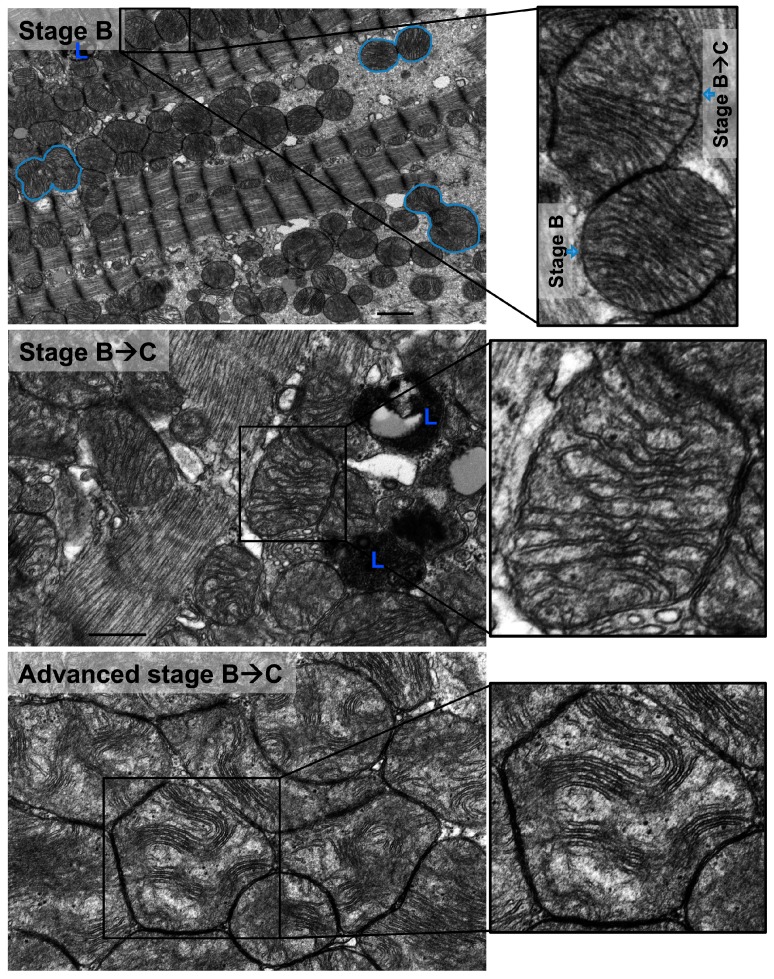
Morphological stages of mitochondrial vacuolar degeneration in a rat model of pressure overload-induced systolic dysfunction. Stage B: there is prominent mitochondrial fission and fragmentation. Blue lines surround mitochondria at early (left middle), intermediate (right lower), and final (right upper) stages of mitochondrial fission. There is focal loss and dissolution of mitochondrial cristae which are now arranged in a horizontal, stacked-up pattern, rather than being dense and folded. Stage B→C: mitochondrial swelling with significant dissolution of mitochondrion cristae at multiple sites. More prominent mitochondrion swelling and dissolution of mitochondrial cristae in advanced stage B→C. The IMM and OMM remain intact. Stage C: Complete dissolution of mitochondrial cristae and IMM. The OMM remains intact. Stage D: Stage C with an associated rupture of OMM (blue arrows pointing at the site of OMM rupture). Stage D^+^: Area(s) where numerous adjacent mitochondria had undergone complete vacuolar degeneration and entire dissolution of mitochondrial cristae, IMM and OMM. L: lysosomes. Photomicrographs are 40k x magnified, scale bar 0.5 μm. In stage B, photomicrograph is 12k × magnified, scale bar 1 μm.

**Figure 6 medicina-55-00239-f006:**
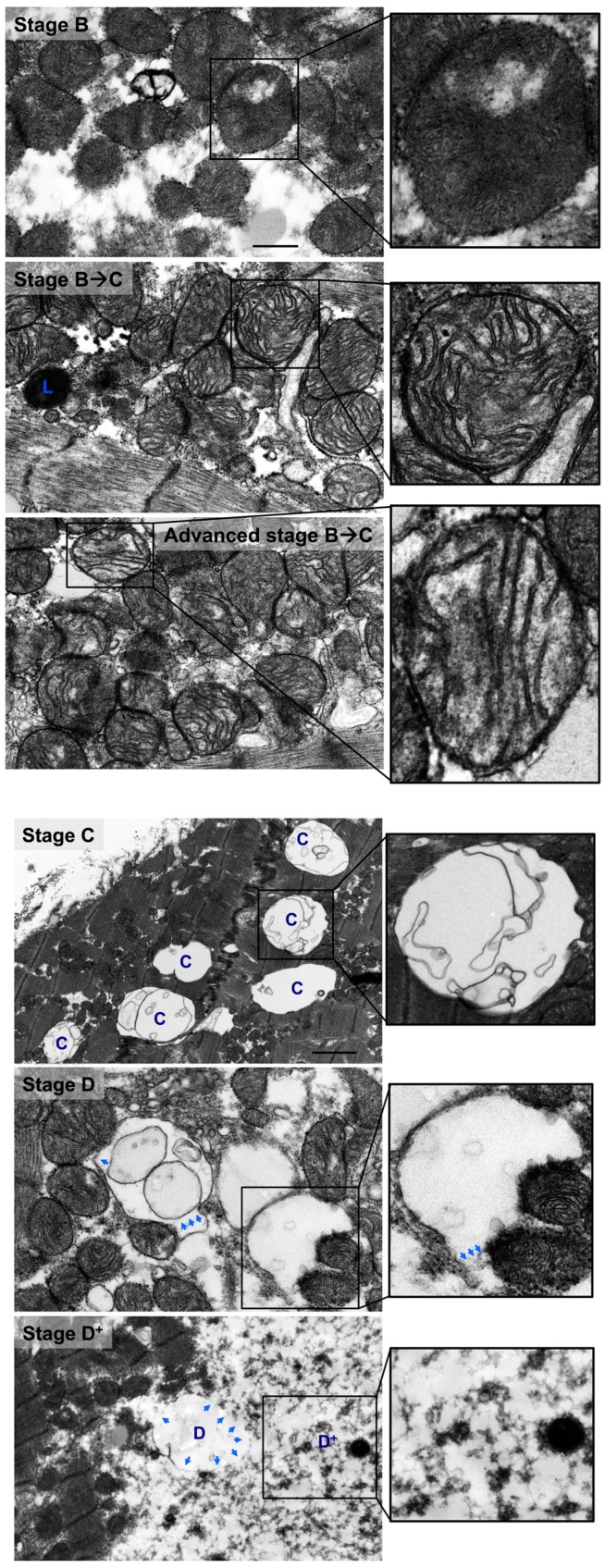
Morphological stages of mitochondrial vacuolar degeneration in human systolic heart failure. Similar pattern of morphological changes of mitochondrial vacuolar degeneration as highlighted in Figure 5 and legend. Blue arrows pointing at sites of OMM rupture. Photomicrographs are 40k × magnified, scale bar 0.5 μm.

**Figure 7 medicina-55-00239-f007:**
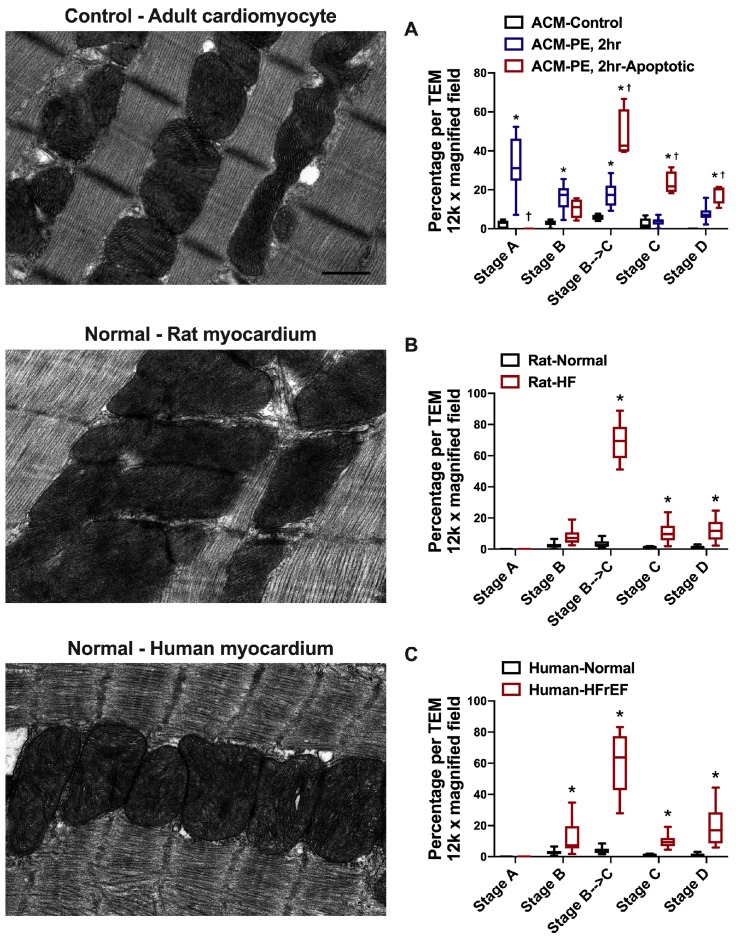
Representative normal transmission electron photomicrographs and quantification of the different stages of mitochondrial vacuolar degeneration in: **A**. Phenylephrine (PE)-stressed adult cardiac myocytes (ACMs). * *p* < 0.05 vs. ACM-Control and ^†^
*p* < 0.05 vs. ACM-PE, 2. **B**. Animal (rat) model of pressure overload-induced HF, and **C**. Human HF with reduced ejection fraction (HFrEF). * *p* < 0.05 vs. Normal. Data are presented as box and whisker plots (lines showing median, 25th and 75th percentiles with whiskers showing minimum and maximum values.

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
