# Peer review of "Morphological Stages of Mitochondrial Vacuolar Degeneration in Phenylephrine-Stressed Cardiac Myocytes and in Animal Models and Human Heart Failure"

_medicina, 2019, doi:10.3390/medicina55060239_

Round 1
Reviewer 1 Report
Please provide the reference for Transmission electron microscopy methods.
Please provide the detailed procedures and information about the rat model (authors may add a different section for it in material and methods).
Also provide the information about the statistical significance of about all the biopsy sample, either from patients or rats. Without statistical significance, how one can interpret the results. Atleast the number of samples should be mentioned. Please justify.
Authors should also add a different section for statistical analysis in material and methods.
What about the control in all the figures. How one can compare the morphological differences?
Author Response
Please provide the reference for Transmission electron microscopy methods.
The reference for transmission electron microscopy has been provided.
Please provide the detailed procedures and information about the rat model (authors may add a different section for it in material and methods).
We did provide a detailed description of the ascending aortic banding procedure in the rat model of pressure overload induced Heart Failure with moderate left ventricular remodeling and systolic dysfunction.
Also provide the information about the statistical significance of about all the biopsy sample, either from patients or rats. Without statistical significance, how one can interpret the results. At least the number of samples should be mentioned. Please justify.
We thank the reviewer for his/her comment. This has been added in the materials and methods section of the manuscript. Lines 152 and 165-166.
Authors should also add a different section for statistical analysis in material and methods.
Now that we have quantified and presented the percentages of the different stages of mitochondrial vacuolar degeneration encountered in Control vs PE-Stressed cardiomyocytes and Normal vs HF; we have added a Statistical analysis section in the materials and methods section of the manuscript.
What about the control in all the figures. How one can compare the morphological differences?
Representative transmission electron photomicrographs from control adult cardiomyocytes and from LV myocardium from rats and patients with normal LV size and function are now presented in Figure 7 of the manuscript.
Reviewer 2 Report
The present study displayed progressive morphological changes of mitochondrial vacuolar degeneration in cardiac myocytes, utilizing transmission electron microscopy, and characterized the variable stages as the indicator to assess the degree of mitochondrial damage in phenylephrine stimulated cardiac myocytes and in myocardium from rat failing heart induced by pressure overload and from patients with HFrEF. Chaanine AH. revealed stages of derangement in mitochondrial morphology in detail according to changes in mitochondrial cristae morphology and mitochondrial vacuolar degeneration. The overall observation proposes a new approach to evaluate the morphological feature in heart failure. However, the following considerations should be assessed with other means to make sure the significance of your histological observations.
Major comments;
1. The significance of changes in mitochondrial cristae morphology and mitochondrial vacuolar degeneration still remains uncertain in mitochondrial dynamics in large part because the author just argued a pathological feature. We understand examining signal transduction pathways is basically out of your scope. But, to state your conclusion in severity of mitochondrial damage, the number of mitochondria, the fusion, and an imbalance in mitochondrial dynamics including mitophagy, apoptosis and energy production should be totally approached not only in histological technique but also in molecular biology ones. Try to examine some parameters by a method except electron microscopy.
2. Checking GTPase and the related proteins in variable stages of mitochondrial cristae morphology would be important to assess pathophysiological function of each stage.
3. It’s unclear at which stage of mitochondrial vacuolar degeneration become irreversible and impairs mitochondrial function. Mitochondria continuously changes the shape through fusion, fission, biogenesis, degradation, and intracellular trafficking to specific locations. Overlapping to comment 2., please discuss pathophysiological features in each determined stage and bring up limitations.
Minor comments;
1. In stage C and D, it would be helpful to clearly visualize the destruction of mitochondrial membranes in multiple fluorescent staining with specific markers for outer mitochondrial membrane, Inner mitochondrial membrane, and inter-membrane space, respectively.
2. In Figure 5, magnification in panel for stage B appears to be different from stage C, but scale bar looks same. Make sure scale bar in each representative panel.
3. Please describe methodology in brief regarding how to isolate cardiac myocytes.
Author Response
The present study displayed progressive morphological changes of mitochondrial vacuolar degeneration in cardiac myocytes, utilizing transmission electron microscopy, and characterized the variable stages as the indicator to assess the degree of mitochondrial damage in phenylephrine stimulated cardiac myocytes and in myocardium from rat failing heart induced by pressure overload and from patients with HFrEF. Chaanine AH, revealed stages of derangement in mitochondrial morphology in detail according to changes in mitochondrial cristae morphology and mitochondrial vacuolar degeneration. The overall observation proposes a new approach to evaluate the morphological feature in heart failure. However, the following considerations should be assessed with other means to make sure the significance of your histological observations.
Major comments;
1. The significance of changes in mitochondrial cristae morphology and mitochondrial vacuolar degeneration still remains uncertain in mitochondrial dynamics in large part because the author just argued a pathological feature. We understand examining signal transduction pathways is basically out of your scope. But, to state your conclusion in severity of mitochondrial damage, the number of mitochondria, the fusion, and an imbalance in mitochondrial dynamics including mitophagy, apoptosis and energy production should be totally approached not only in histological technique but also in molecular biology ones. Try to examine some parameters by a method except electron microscopy.
We thank the reviewer for his/her comment. We have already addressed and published above aforementioned concerns previously. In our previous work we have studied autophagic flux and markers of autophagy in PE-stressed cardiomyocytes for 2 hrs in vitro (Chaanine et al, Cell Death and Disease 2012). Also, we have studied apoptosis in PE-stressed cardiomyocytes at different time points; 2hr, 4hr, 6hr and 12 hr after PE stressor (Chaanine et al, AJP Heart and Circulatory Physiology 2016). We showed that autophagy is upregulated and mitophagy markers are increased in PE-stressed ACM for two hours and then decline 4 hrs after initiation of PE stress, whereas, there is exponential increases in apoptosis 4, 6 and 12 hrs after initiation of PE stress. Markers of autophagy and apoptosis are also studied, in vivo, in the rat model of PO induced HF (Chaanine et al, Cell Death and Disease 2012, Circulation: HF 2013 and JAHA 2017).
We looked at mitochondrial dynamics and oxidative phosphorylation as well as oxidative capacity in the rat model of PO induced HF and in Human HFrEF (Chaanine et al, AJP Heart and Circulatory Physiology 2016 and Circulation: HF 2019), respectively. We found evidence of DRP-1 dephosphorylation at S637 in rat HF, while DRP-1 expression was upregulated in Human HFrEF. There was no significant changes in the expression of MFN2 and OpA-1. Oxidative capacity was impaired in rat-HF and in Human HFrEF.
2. Checking GTPase and the related proteins in variable stages of mitochondrial cristae morphology would be important to assess pathophysiological function of each stage.
We thank the reviewer for his/her comment. Although OPA-1 expression is unchanged between Normal and HF, we have not yet assessed OPA-1 activity under these conditions. We plan to do this experiment in a future study. In this study we focused on describing and presenting the different stages of mitochondrial vacuolar degeneration.
3. It’s unclear at which stage of mitochondrial vacuolar degeneration become irreversible and impairs mitochondrial function. Mitochondria continuously changes the shape through fusion, fission, biogenesis, degradation, and intracellular trafficking to specific locations. Overlapping to comment 2., please discuss pathophysiological features in each determined stage and bring up limitations.
We have addressed this in the discussion section. Based on the quantified data in figure 7 and from our prior data in systolic HF (mitochondrial dysfunction and apoptosis) we speculate that mitochondrial dysfunction and apoptosis become significant when about 60-70% of the mitochondria on average are at stage BàC (early and advanced), with 10-20% being at Stages C and D. Of course, this remains an observation and has to be proven and validated in a subsequent study. Data in systolic HF, follow a similar trend seen in PE-apoptotic cardiomyocytes.
Minor comments;
1. In stage C and D, it would be helpful to clearly visualize the destruction of mitochondrial membranes in multiple fluorescent staining with specific markers for outer mitochondrial membrane, Inner mitochondrial membrane, and inter-membrane space, respectively.
We apologize it is not feasible to perform this experiment at this moment.
2. In Figure 5, magnification in panel for stage B appears to be different from stage C, but scale bar looks same. Make sure scale bar in each representative panel.
We thank the reviewer for highlighting this error. Indeed the Stage B TEM image in Figure 5 is a 12k x magnified image while the rest of the TEM images are 40k x magnified. We have changed the scale bar and corrected the figure legend.
3. Please describe methodology in brief regarding how to isolate cardiac myocytes.
We have done so in the materials and Methods section.
Reviewer 3 Report
The study by Chaanine examines mitochondrial morphology through transmission electron microscopy using different experimental systems (in vitro and in vivo) of systolic HF. The author describes progressive stages of mitochondrial vacuolar degeneration, defining morphologically the severity of mitochondrial damage.
- It would be helpful if the author provides control images of cardiac myocytes, ie non-stressed, in order to make more clear to the reader the observed morphological mitochondrial alterations.
-Letters and annotations within the images are not always well visible (eg in Figure 1, Figure 5). Maybe a different coloring system should be chosen instead of blue.
-The number of samples used in this study is unclear
-The in vitro system used in this study is not clear, what experimental animal system was used to isolate cardiac myocytes? The reader is referred to a reference in the methods section but I think it will be important to clarify this in the results.
-What is the rationale behind the use of BNIP3 or FOXO3a adenoviral transfected cells? Is this just an additional experimental system to demonstrate or confirm the observed mitochondrial morphological changes? Please clarify in Results section.
-In what proportion within a sample are observed the different stages of mitochondrial degeneration? Is there any way of quantifying mitochondria at stage A, B-C, C or D?
-It would be helpful if the author provides an overall schematic diagram showing mitochondrial morphological changes in the different stages of vacuolar differentiation
-The overall scope of this study is somehow unclear to me
Author Response
The study by Chaanine examines mitochondrial morphology through transmission electron microscopy using different experimental systems (in vitro and in vivo) of systolic HF. The author describes progressive stages of mitochondrial vacuolar degeneration, defining morphologically the severity of mitochondrial damage.
- It would be helpful if the author provides control images of cardiac myocytes, ie non-stressed, in order to make more clear to the reader the observed morphological mitochondrial alterations.
As suggested, we have provided control images presented in Figure 7.
- Letters and annotations within the images are not always well visible (eg in Figure 1, Figure 5). Maybe a different coloring system should be chosen instead of blue.
We have changed the figures according to the reviewer’s suggestions and provided figures with higher resolution.
- The number of samples used in this study is unclear
We clarified this in the Materials and Methods section of the manuscript.
- The in vitro system used in this study is not clear, what experimental animal system was used to isolate cardiac myocytes? The reader is referred to a reference in the methods section but I think it will be important to clarify this in the results.
This has been clarified in the materials and methods section of the manuscript.
- What is the rationale behind the use of BNIP3 or FOXO3a adenoviral transfected cells? Is this just an additional experimental system to demonstrate or confirm the observed mitochondrial morphological changes? Please clarify in Results section.
We thank the reviewer for his suggestion. We have clarified this in the results section, lines 248-259.
- In what proportion within a sample are observed the different stages of mitochondrial degeneration? Is there any way of quantifying mitochondria at stage A, B-C, C or D?
We have quantified the different stages in 12k x magnified images and presented the results as percentage of total mitochondria per image/field. This is presented in Figure 7. The quantification was performed in at least 3 different samples per group (except in Human-Control, n=2 samples) and at least 4 images were quantified per sample.
- It would be helpful if the author provides an overall schematic diagram showing mitochondrial morphological changes in the different stages of vacuolar differentiation
The zoomed TEM images clearly demonstrate that.
- The overall scope of this study is somehow unclear to me
We acknowledge limitations to the study, but now that we have quantified the different stages of mitochondrial vacuolar degeneration and in correlation we prior data that we have published, we observe that when about at least 60-70% of the mitochondria are at stage BàC of vacuolar degeneration that mitochondrial dysfunction and apoptosis take place as seen in systolic HF. This has been discussed in the results and discussion section of the manuscript. Of course this is an observation and need to be validated in future study.
Round 2
Reviewer 1 Report
Thanks for updates.
Reviewer 2 Report
The author tried to address reviewer’s comments mostly by referencing previous reports. However, a main subject regarding physiological significance of each stage related to mitochondrial morphological changes is fundamentally unsolved. It’s not still worthy of acceptance unless any physiological significance is made sure in your histological observation and classification.